# The Use of Mesenchymal Stem Cells in the Complex Treatment of Kidney Tuberculosis (Experimental Study)

**DOI:** 10.3390/biomedicines10123062

**Published:** 2022-11-28

**Authors:** Alexander N. Muraviov, Tatiana I. Vinogradova, Anna N. Remezova, Boris M. Ariel, Anna A. Gorelova, Nadezhda V. Orlova, Natalia M. Yudintceva, Diljara S. Esmedliaeva, Marina E. Dyakova, Marine Z. Dogonadze, Natalia V. Zabolotnykh, Irina A. Garapach, Olga S. Maslak, Yuri A. Kirillov, Sergei E. Timofeev, Yulia S. Krylova, Petr K. Yablonskiy

**Affiliations:** 1Saint-Petersburg State Research Institute of Phthisiopulmonology of the Ministry of Health of the Russian Federation, 191036 Saint-Petersburg, Russia; 2Institute of Cytology of Russian Academy of Sciences, Tikhoretsky 4, 194064 Saint-Petersburg, Russia; 3Avtsyn Research Institute of Human Morphology of Federal State Budgetary Scientific Institution “Petrovsky National Research Centre of Surgery”, 119991 Moscow, Russia; 4Moscow Multidisciplinary Clinical Center “Kommunarka” of the Moscow Health Department, 108814 Moscow, Russia

**Keywords:** experimental nephrotuberculosis, MSC, reparative reaction

## Abstract

In recent years, the application of mesenchymal stem cells (MSCs) has been recognized as a promising method for treatment of different diseases associated with inflammation and sclerosis, which include nephrotuberculosis. The aim of our study is to investigate the effectiveness of MSCs in the complex therapy of experimental rabbit kidney tuberculosis and to evaluate the effect of cell therapy on the reparative processes. Methods: To simulate kidney tuberculosis, a suspension of the standard strain Mycobacterium tuberculosis H37Rv (10^6^ CFU) was used, which was injected into the cortical layer of the lower pole parenchyma of the left kidney under ultrasound control in rabbits. Anti-tuberculosis therapy (aTBT) was started on the 18th day after infection. MSCs (5 × 10^7^ cells) were transplanted intravenously after the start of aTBT. Results: 2.5 months after infection, all animals showed renal failure. Conducted aTBT significantly reduced the level of albumin, ceruloplasmin, elastase and the severity of disorders in the proteinase/inhibitor system and increased the productive nature of inflammation. A month after MSC transplantation, the level of inflammatory reaction activity proteins decreased, the area of specific and destructive inflammation in kidneys decreased and the formation of mature connective tissue was noted, which indicates the reparative reaction activation.

## 1. Introduction

According to the World Health Organization, 10 million annual cases of tuberculosis are diagnosed worldwide; among them, extrapulmonary localizations range from 5% to 45% with urogenital tuberculosis being a significant part of them [1]. It poses a much lower epidemiological risk than respiratory tuberculosis and is diagnosed relatively late, when in more than half of the cases, patients have already developed certain complications and irreversible changes [2,3]. One of them is renal failure, since pathomorphological changes in the kidneys during infection in 54.5–84.7% of cases show irreversible parenchymal injury, accompanied by the formation of ureteral strictures, followed by hydronephrosis, which is treated mostly with reconstructive plastic surgery [3,4].

Fibrotic processes, typical of a long course of specific inflammation and its chronicity, contribute partially to ureterohydronephrotic transformation of the kidney, along with the formation of granulomas and their calcification. They are inevitable, on the one hand, due to the late diagnosis of nephrotuberculosis as mentioned above and, on the other hand, due to the nephrotoxic effect of anti-tubercular drugs [5].

In recent years, the latest biomedical technologies have been increasingly used to restore the structure and function of damaged organs and tissues, in particular cell therapy based on the transplantation of mesenchymal stem cells into the affected organ. Mesenchymal stem cells (MSCs) present one of the main reparation tools in different pathological processes. These pluripotent cells have antimicrobial properties [6,7], outstanding regenerative potential due to their unique ability to migrate to the site of injury, persist there and secrete a wide range of bioactive macromolecules that regulate the damaged tissue restoration [8,9,10], differentiate into different cell types [11,12,13] and have an immunomodulatory effect [14,15]. In other words, theoretically, stimulation of tissue regeneration with MSCs in various pathologies as one of the cell therapy methods, opens up a broad therapeutic perspective. In addition, the caution which theorists and clinicians sometimes exercise thereof is undeniably justified. It goes without saying that before the introduction of cell therapy into the process, it is necessary to create standards that can significantly reduce the risk of its use in the clinic.

MSCs are currently believed to participate in regulation of tissue regeneration (even at a low level of engraftment) by the recruitment of cells to the damaged area by chemotaxis, and paracrine regulation of the behavior of cells in the microenvironment [16,17,18].

The MSC secretome includes factors responsible for angio- and neurogenesis, cell proliferation, viability and chemotaxis [19,20,21,22]. However, paracrine signaling is not limited to certain growth factors and cytokines; extracellular vesicles (EVs) secreted by MSCs play the most important role here [22,23].

The use of MSCs and their derivatives in the form of extracellular vesicles (MSC-EVs) is recognized as a promising treatment option in urology [24]. Positive results have been obtained in preclinical studies of acute and chronic renal failure, early diabetic nephropathy and bladder reconstruction [25,26,27,28,29]. In a model of chronic renal failure in diabetic rats, the protective effect of intravenous MSC administration on renal function, expressed in a decreased level of pro-inflammatory cytokines (IL-6, IL-1β and TNF-α) and profibrotic factor (TGF-β) in renal tissue and blood plasma, was revealed. At the same time, a decrease in the inflammatory process activity was observed in general, including its final stages, characterized by the development of fibrosis [28].

In nephrotuberculosis, which results in severe fibrotic changes in the urinary tract, up to the hydronephrotic transformation of the kidneys, MSC transplantation can potentially have a significant impact on the inflammatory response and the severity of repair. Earlier we demonstrated that intravenous administration of MSCs results in accumulation and retention of MSCs in rabbit kidney tissue affected by mycobacteria [30].

Our first studies to evaluate the effectiveness of MSCs in the activation of reparative processes on the bladder and female genitalia TB model showed positive results [31,32,33]. The objective of this study is to investigate the effectiveness of MSCs in the complex treatment of experimental rabbit kidney TB and to evaluate the effect of cell therapy on the nature of reparative processes.

## 2. Materials and Methods

### 2.1. Laboratory Animals

The study included the results of a dynamic follow-up of 20 (Figure 1) chinchilla rabbits weighing 3329 ± 272.6 g (the animals were received from the FSUE Nursery of laboratory animals “Rappolovo” of the Federal State Budgetary Institution National Centre “Kurchatov Institute”, Leningradskaya oblast’, Russia), which were kept in a certified vivarium on the basis of the Federal State Budgetary Institution “SPb NRIF” of the Ministry of Health of Russia, in accordance with GOST 33216-2014 “Rules for working with laboratory rodents and rabbits”. The permission of the Independent Ethics Committee of the Federal State Budgetary Institution “SPb NRIF” of the Ministry of Health of Russia to conduct this study was obtained.

### 2.2. Isolation, Cultivation and Labeling of MSCs

The bone marrow was collected from healthy chinchilla rabbits (*n* = 4). For all four rabbits, 8 mL of bone marrow was aspirated aseptically from the iliac crest: 4 vacutainers (volume 2 mL) with lithium heparin (BD Biosciences, Le Pont de Claix, France) were taken from each animal [34,35,36,37]. MSCs were obtained using the Ficoll gradient method. The bone marrow was diluted (1:1) with phosphate-buffered saline (PBS) and transferred into 50 mL conical centrifuge tubes (TPP, Trasadingen, Switzerland). SetMate (Stemcell Technologies, Kent, WA, USA) and Ficoll (Merck, Rahway, NJ, USA) tubes were used for cell isolation. The bone marrow was gently added to the Ficoll surface with a proportion of 2:1. The centrifugation was performed for 10 min at 1200× *g*. After centrifugation, plasma with mononuclear cells was carefully transferred into a new 50 mL centrifugation tube and washed twice with 10 mL PBS. After washing, cells were counted and plated into the dish (Nunc, Rochester, NY, USA) at a density of 5 × 10^3^/cm^2^. Cells were cultivated under standard conditions in a DMEM/F12 medium containing 10% fetal bovine serum and gentamicin solution (Gibco, Oxford, UK) 50 µg/mL. Cells of the 4th-5th passages were used in the experiments. Immunophenotyping of cells was performed using Abcam monoclonal antibodies (USA) on an EpicsXL flow cytometer (BeckmanCoulter, Brea, CA, USA).

For stem cell tracking, MSCs were labeled with superparamagnetic iron oxide nanoparticles (SPIONs). MSCs were incubated with SPIONs at a concentration of 150 μg/mL for 24 h in a CO_2_ incubator. Cell viability was assessed using 0.4% Trypan blue dye (Biolot, Saint Petersburg, Russia). Cytotoxicity of nanoparticles and the metabolic activity of cells (MTT method) were analyzed using a colorimetric assay (Vybrant^®^ Life Technologies, Carlsbad, CA, USA) in accordance with the manufacturer’s protocol [32,33].

### 2.3. Animal Challenge

The drug-susceptible reference strain Mycobacterium tuberculosis H37Rv (TBC # 1/47, source—Institute of Hygiene and Epidemiology, Prague, 1976) obtained from the FSBI “Scientific Centre for Expert Evaluation of Medicinal Products” of the Ministry of Health of Russia was used as an infectious agent. To simulate kidney tuberculosis under ultrasound control, a fine-needle puncture of the cortical layer of the lower pole parenchyma of the left kidney was performed in the rabbits, and a suspension of M. tuberculosis H37R (Mtb) reference strain at a dose of 10^6^ CFU in 0.2 mL of saline solution was injected [38]. For anesthesia, intramuscular administration of 1.0–1.5 mL of Zoletil (zolazepam + tiletamine (Virbak SA, Carros, France) at a dose of 25 mg/kg of body weight) and Xila (Xylazine, Interchemie Werken “de Adelaar” BV, Netherlands, as 1.0–1.5 mL of 2% solution) were used. Within 5 days of the infection, antibiotic prophylaxis was administered (cefazolin at a dose of 50,000 U/kg, 1.5 mL intramuscularly, JSC Pharmasyntez, Irkutsk, Russia).

The development of nephrotuberculosis was monitored 18 days after the infection using a delayed-type hypersensitivity test with the recombinant tuberculosis allergen “Diaskintest”^®^ (Generium, Moscow, Russia), performing the abdominal computed tomography scan on a Toshiba One Aquilion tomograph.

Diaskintest^®^ was injected at a concentration of 2 μg/mL in 0.1 mL of saline intradermally on the back in the kidney projection area. Computed tomography was performed by bolus injection of a contrast agent Ultravist-370 (4 mL at a rate of 1 mL/min) into the marginal ear vein. Then, within 30 s, sequential images of the abdomen were obtained and served as the basis for the perfusion map.

All infected rabbits were divided into 3 groups using an online random number generator: group 1—challenge control (untreated rabbits); group 2—rabbits with positive results of Diaskintest, treated orally for 3 months with isoniazid (10 mg/kg, JSC “Moskhimfarmpreparaty named after N.A. Semashko”, Russia); pyrazinamide (15 mg/kg, JSC Pharma sintez, Irkutsk, Russia); ethambutol (20 mg/kg, Shreya Life Sciences, India). Anti-tubercular therapy was started 18 days after the challenge. After 3 months, the animals were euthanized by an overdose of anesthetic agents injected into the marginal ear vein: sodium thiopental 250 mg (Sintez, Kurgan, Russia) and pipecuronium bromide 1 mg (Veropharm, Moscow, Russia); group 3—rabbits, which were treated with a suspension of 5 × 10^7^ MSC in 2 mL of PBS injected into the lateral vein of the left ear 2 months after the start of the same anti-tubercular therapy. One month after MSC transplantation, the animals were euthanized in the same way as the rabbits in the 2nd group.

### 2.4. Infection Severity

Infection severity was evaluated based on:

(1) changes of body weight over time (weekly); (2) biochemical indicators of the general inflammatory response and the functional state of the kidneys (baseline, 18 days after the challenge, 2 and 3 months after the start of treatment): the activity of purine metabolism enzymes—the total activity of adenosine deaminase (ADA) and its isoenzymes (ecto-ADA-1 and ecto-ADA-2)—using the method of G. Giusti (1974); concentration of ceruloplasmin (CP) using the method of Ravin; creatinine (CR) and albumin (AL) concentrations were determined with Beckman Coulter reagents on a Synchron CX5 PRO biochemical instrument (Beckman Coulter, USA); markers of destruction and remodeling: the activity of serine protease elastase (EL) using the method of L. Visser and E. R. Blout (1972) [39] and a concentration of metalloproteinase-1 (MMP-1), metalloproteinase-3 (MMP-3) and their tissue inhibitor TIMP-1 using the ELISA method according to the manufacturer’s protocol (Cusabio, China); (3) data of histological and morphometric examination of the kidneys (3 months after the infection).

The kidneys were fixed entirely in 10% neutral formalin and embedded in paraffin, 4–6 µm thick sections were stained with hematoxylin and eosin, according to the methods of Van Gieson, Ziehl-Neelsen and Perls (to detect iron). Slides were scanned with a Leica Aperio AT2 scanner (Leica Biosystems, Germany; PlanApo 20×; numerical aperture 0.75 and analyzed using Aperio ImageScope software.

Preview and selection of areas for analysis were performed using QuPathv0.2.3 software (University of Edinburgh, UK) [40]. For morphometry, the NIH ImageJ version 1.52a (National Institutes of Health, USA) with additional plug-ins was used.

Areas for measurement were selected in digital images of histological slides. The selected areas from the upper, middle and lower poles of the kidneys were polygonal in shape, completely capturing the cortical and medullar parenchyma from the capsule to the pelvis; their perimeter was 22–31 k sq. µm.

MSCs containing the dye were detected in tissue by indirect immunofluorescence analysis. To do this, the tissue was embedded in a Tissue-Tek solution ((Sakura Finetek Europe BV, Alphenanden Rijn, The Netherlands) and frozen in liquid nitrogen by repeated immersion. The 10–15 µm thick cryotomy sections were placed on SuperFrost glass slides and fixed with 10% neutral formalin solution. For nuclear staining, sections were immersed for 5 min in 1% DAPI dye (4,6-diamidino-2-phenylindole dihydrochloride), washed three times with saline PBS, and mounted in a Mountingmedium medium (Pharmacia Biotech, Uppsala, Sweden). Stained cells were identified using a LeicaTCSSL confocal microscope (Zeiss, Oberkochen, Germany) and an argon laser with a wavelength of 488 nm.

For the morphological and functional characterization of the kidneys, a morphometric study of the glomeruli (determination of the area of the Shumlyansky–Bowman capsule inner surface, the area of capsular space and the area occupied by capillaries) was performed; the measurement results were interpreted as an assessment of the filtration capacity of the kidneys. In addition, the height of the epithelial cells of the proximal and distal convoluted tubules, and also the collecting tubules, were measured in order to study water and electrolyte reabsorption and renal output efficiency through the collecting ducts. The area of necrotic lesions and the area of fibrosis were assessed.

### 2.5. Statistical Processing

The results of the morphometric study were evaluated using specialized software: R-4.0.4 (The R Foundation, Vienna, Austria) and RStudio Desktop (Version 1.3.1093, RStudio Inc., Boston, MA, USA). We used the Statistica 7.0 software package (StatSoftInc., Tulsa, OK, USA). The nature of the sample data distribution was determined, and in case of deviation from the normal distribution (according to the Shapiro–Wilk criterion), the median (Me) and the first and third quartiles (Q1-Q3) were calculated. Significance of differences was assessed using the Mann–Whitney U-test and the Kruskal–Wallace test. Correlations were determined by calculating and evaluating the Spearman coefficient.

## 3. Results

### 3.1. Immunodiagnostics of Nephrotuberculosis

All challenged animals developed a specific inflammatory process in the kidneys with a characteristic immunological reorganization, which was confirmed by erythema of 15.3 ± 2.8 mm in size in response to Diaskintest^®^ administration 18 days after the challenge and its absence (Figure 2a,b, respectively). According to the results of the abdominal computed tomography scan 30 days after the challenge, a clearly defined area of reduced perfusion was revealed in the left (infected) kidney (Figure 2c). During standard computed tomography, a destruction focus is visualized in this area (Figure 2d).

### 3.2. Examination of Nanoparticle-Labeled MSCs

Reflective laser scanning with confocal microscopy demonstrated internalization of nanoparticles in the cytoplasm (Figure 3a). SPIONs are visualized as red dots present in the cytosol but not in the nucleus. MSCs without SPIONs were used as an internal negative control (Figure 3b).

### 3.3. Evaluation of Biochemical Indicators of the General Inflammatory Response and the Functional State of the Kidneys

Assessment of the state of the kidneys by creatinine level with the inflammatory response severity was compared according to the level AL, CP, markers of purine metabolism in combination with markers of destruction and remodeling—EL, and indicators of the MMP/TIMP-1 system (Table 1).

No significant changes in the CR, AL and CP levels were found in any of the groups by the 18th day after the challenge, compared with the baseline. At the same time, also during this period, an increase in total ADA activity was observed due to ecto-ADA-1, while ecto-ADA-2 remained unchanged.

In addition, an increase in EL activity (*p* = 0.049) and a decrease in TIMP-1 level (*p* = 0.03) were observed, while the MMP values remained unchanged. Changes in Fi were unidirectionally associated with the total ADA activity and ecto-ADA-1 activity (r = 0.61; *p* = 0.002; r = 0.62; *p* = 0.0015) and oppositely with AL (r = −0.44; *p* = 0.03, respectively). A decrease in AL level was seen with an increase in TIMP-1, as far can be judged by the correlation coefficient (r = −0.47; *p* = 0.03). Changes in MMF and EL were unidirectional. The dependence of changes between the levels of metalloproteinasesMMP-1 and MMP-3 (r = 0.47, *p* = 0.04), as well as between one of the metalloproteinases (MMP-1) and serine proteinase EL (r = 0.51, *p* = 0.01) were revealed. This is not surprising, since the various characteristics of the inflammatory process are, generally speaking, interdependent.

Two months after the challenge, the first group of rabbits showed a significant increase in creatinine (CR) level compared with the baseline (*p* = 0.019), while AL (albumin) and Fi remained the same. An increase in total ADA activity (4-fold), ecto-ADA-1 (3.8-fold) and a tendency towards an increase in ecto-ADA-2 were revealed.

The TIMP-1 level significantly changed from baseline; however, values of EL, MMP-1 and MMP-3 were quite stable, which suggests an imbalance in the proteinase/inhibitor system.

After 3 months, the first group of the rabbits still had elevated creatinine (CR), total ADA and ADA-1 activity levels, although these values were slightly lower compared to the previous study period; by this time, a significant decrease in MMP-1 was also observed (*p* = 0.03). Other indicators remained unchanged relative to the previous observation periods. An increase in creatinine (CR) level was associated with an increase in total ADA and ecto-ADA-1 activity (r = 0.85; *p* = 0.035; r = 0.9; *p* = 0.04, respectively), which indicates the development of an inflammatory process in combination with a decrease in kidney function during this observation period.

In the second group of rabbits, after a two-month course of anti-tubercular treatment, creatinine (CR) remained elevated, while the Fi level decreased and significantly differed from that in the first group (*p* = 0.008). A high total of ADA and ecto-ADA-1 activity and no changes in ecto-ADA-2 were noted. Changes in total ADA activity were combined with an increase in CR (r = 0.45; *p* = 0.04). No significant changes in TIMP-1, EL or MMP were observed, which can be regarded as the absence of severe disturbances in the proteinase/inhibitor system.

In the third group of rabbits, which, along with chemotherapy, were injected with MSCs, changes in biochemical indicators were the same as in the animals in the second group, reflecting the presence of an inflammatory process with persistence of renal failure, but less severe than in other groups.

For this group of animals, the increase in the CR level was the lowest and significantly differed from that in the first group (*p* = 0.003). Fi was also lower than in the first group (*p* = 0.01). When recalculating the concentrations of MMP/inhibitors system indicators according to the creatinine level in this group, the highest level of TIMP-1 (1.31 vs. 0.003) in the first group (*p* = 0.03) and the lowest level of MMP-1 (0.005 vs. 0.003) in the first group (*p* = 0.03) were noted (Figure 3). The same correlations remained between Fi and TIMP-1 (r = −0.45; *p* = 0.04) on the one hand and between MMP-1 and MMP-3 (r = 0.54; *p* = 0.01) on the other hand. The obtained results suggest that MSCs in combination with anti-tubercular drugs led to a less intense activity of the inflammatory response and to an acceleration of the reparative reaction.

### 3.4. Histological and Morphometric Study

Histological examination of the kidneys in the first group of rabbits revealed acute inflammatory changes with hyperemia and an abundance of hemorrhages and thrombosis of the parenchyma of both kidneys had necrotic lesions surrounded by pseudoeosinophilic leukocytes, macrophages, plasmocytes, lymphocytes, epithelioid cells, as well as Langhans giant multinucleated cells of various degree of maturity. In addition, in some necrotic lesions, decay areas were determined with a tendency to form three-layer wall cavities. Ziehl-Neelsen staining revealed numerous acid-fast mycobacteria. Statistical analysis did not reveal significant differences in the area of necrosis between the groups.

In the second group of rabbits, who received a course of anti-tubercular chemotherapy, specific inflammation was noted in the kidneys with a predominance of a productive reaction revealed by massive macrophage and lymphocytic infiltration with single giant cells, and necrotic lesions, much smaller than in the first group of rabbits. Ziehl-Neelsen staining revealed single acid-fast rods in the areas of necrosis. Van Gieson staining demonstrated the initial signs of fibrosis, while no mature connective tissue was detected.

In the third group of rabbits (with MSC transplantation secondary to the chemotherapy), the morphological pattern was characterized by severe angiomatosis and massive macrophage-lymphocytic infiltration. Van Gieson staining found the formation of mature fibrous connective tissue at the site of necrotic lesions. Ziehl-Neelsen staining determined acid-fast rods. Perls staining revealed focal accumulations of macrophages with an abundance of coarse grains of hemosiderin and small cells with a slightly elongated nucleus, located both in the stroma, between collagen fibers, and in the walls of small vessels and containing fine-grained inclusions. Small cells with elongated nuclei were assumed to be mesenchymal stem cells. Using the method of confocal microscopy of cryosections of kidney tissue samples, cells labeled with nanoparticles (Figure 4) were detected. Based on the literature review, as well as our previous data, it can be assumed that these are MSCs located in the tissue of the affected kidney.

Thus, MSCs in the etiotropic therapy of experimental nephrotuberculosis led to a decrease in the distribution of specific inflammation in the kidneys, a decrease in its severity and an acceleration of the reparative reaction with the formation of mature connective tissue (Figure 5 and Figure 6).

A morphometric study of such indicators as the area of the Shumlyansky capsule inner surface, the area of the capsular space and the area occupied by the glomerular capillaries revealed similar changes in the left (infected) and right (control) kidneys, both in qualitative and quantitative terms, and the statistical processing of morphometric data showed that the differences were not significant (*p* < 0.05) (Figure 7 and Figure 8). This made it possible to combine the samples related to the left and right kidneys, and to consider the renal function as a whole in rabbits of each group as its generalized assessment.

Obviously, these measurements demonstrate that in all experimental rabbits the numerical values of glomerular parameters are 1.4–2.4-fold higher than in the control group (in healthy animals), which becomes especially clear if representing this as a ratio to the corresponding values in healthy animals taken as a unit.

A morphometric study of such indicators as the height of epithelial cells of the proximal and distal convoluted tubules and collecting tubes also revealed statistically significant differences (*p* > 0.05) between the left infected kidney and the right control, where, according to histological examination, specific inflammatory changes were found as well. It must be assumed that, in contrast to glomerular filtration, the processes of reabsorption in the left and right kidneys, as well as the excretion of urine through the collecting tubules, differed from each other. In this regard, the samples related to the left and right kidneys were considered in a separate way.

Obviously, these measurements demonstrate that in almost all the experimental rabbits, with the exception of one, the numerical values of the heights of epithelial cells of the convoluted tubules and collecting ducts are 1.1–1.4-fold higher than in the control group (in healthy animals), which becomes especially clear if represented as a ratio to the corresponding values in healthy animals, taken as a unit.

## 4. Discussion

Our experiments showed that when a suspension of M. tuberculosis H37Rv is injected into the cortical layer of the lower pole of the left kidney, nephrotuberculosis naturally develops in rabbits, which is confirmed by the data of ultrasound, microbiological and histological studies and the results of Diaskintest. The histological method also revealed a specific lesion of another, non-operated kidney.

A biochemical study assured us that an acute inflammatory process with a typical decrease in albumin level, and an increase in Fi as one of the reactants of the acute phase of inflammation, was accompanied by the development of renal failure with an increase in blood plasma creatinine. This was combined with characteristic disorders of purine metabolism (an increase in total ADA and ecto-ADA-1 activity, an imbalance in the system of proteinases and their inhibitors (respectively, a decrease in TIMP-1), and an increase in EL activity). A decrease in albumin level in combination with an increase in TIMP-1 and EL activity, along with the direct correlation between MMP activity, or, more precisely, between tMMP-1 and MMP-3 levels, indicates the activation of proteolysis processes. According to the literature, an imbalance of proteinases and their inhibitors reflects the activity of inflammation and the progression of nephrosclerosis [41]; MMPs can damage tubular cells as well activate a number of pro-inflammatory and profibrotic signals in the kidneys, which ultimately lead to the progression of chronic renal failure [42,43].

Other relationships between indicators characterizing different aspects of the inflammatory response were also identified. Thus, it is important to note the inverse correlation between albumin levels and elastase activity, since albumin is known to selectively and reversibly inhibit the TNF-induced fall in cAMP levels [44], and EL secretion is controlled by adenosine [45].

Adenosine is an important regulatory molecule that increases rapidly with inflammation. The level of adenosine depends on the ADA-1 isoenzyme activity, in particular, the high-affinity A1. Adenosine reduces the glomerular filtration rate by constriction of afferent arterioles, especially in superficial nephrons, which is consistent with the concept of metabolic control of organ function [44,45].

In this aspect, the meaning of the most important results of morphometry, in particular, the increase in the area occupied by capillaries in the glomeruli, becomes clear. Indeed, the decrease in the glomerular filtration rate under the influence of adenosine is, so to speak, compensated by an increase in the total area of the filtering surface of the capillaries. In this regard, an excessive amount of filtrate accumulates in the glomeruli compared to normal, which in turn leads to stretching of the capsular space and expansion of the area of the Shumlyansky–Bowman capsule inner surface.

Obviously, excessive accumulation of filtrate in the capsular space also requires compensation. It occurs, apparently, due to an increase in reabsorption in the convoluted tubules. According to our morphometric estimates, which are quite significant, the height of the epithelial cells of the convoluted tubules increased in rabbits, and this can be considered as a compensatory response to the relative retention of urine in the upper parts of the nephron. Similarly, we consider morphometric data regarding the collecting tubules.

Thus, our observations showed that in all rabbits, regardless of their group affiliation, a high activity of the inflammatory response remained throughout the experiment. At the same time, in the third group of rabbits, which, along with anti-tubercular drugs, were treated with MSCs, it was minimal. Specifically in the animals of this group, according to the histological examination, initial manifestations of the reparative reaction were noted.

All this leads to the conclusion that combined therapy of nephrotuberculosis in rabbits consisting of MSC administration in combination with a standard anti-tubercular regimen can be considered quite effective. There is no doubt about the feasibility of its use in the near future in the treatment of nephrotuberculosis in humans, naturally, after the necessary clinical trials have been carried out.

## 5. Conclusions

The introduction of a suspension of M. tuberculosis H37Rv into the cortical layer of the lower pole of the left kidney in rabbits led to the development of specific inflammatory changes in both kidneys and the development of a general inflammatory reaction that remained highly active throughout the experiment, despite the therapy.All challenged animals developed renal failure, characterized by an increase in the creatinine level, total ADA activity and ecto-ADA-1 activity.Changes in biochemical indicators in the second and third group of rabbits, which were of a similar nature, support the fact that the use of MSCs in combination with anti-tubercular drugs reduces the activity of the inflammatory response and contributes to its stabilization.Specific inflammation in the kidneys of the second and third group rabbits was of a productive nature; their necrotic lesions were much smaller than in the first group rabbits.There were also differences in the development of fibrotic changes, which were absent in the first group rabbits, only outlined in the second group rabbits and clearly expressed in the third group rabbits.Renal failure in nephrotuberculosis contributed to the development of a compensatory reaction from the glomeruli, convoluted tubules and collecting tubules, the severity of which was determined by a morphometric study.MSCs in the etiotropic therapy of experimental nephrotuberculosis led to a decrease in the extent of specific inflammation distribution in the kidneys, a decrease in its activity and an acceleration of the reparative reaction with mature connective tissue formation.

## Figures and Tables

**Figure 1 biomedicines-10-03062-f001:**
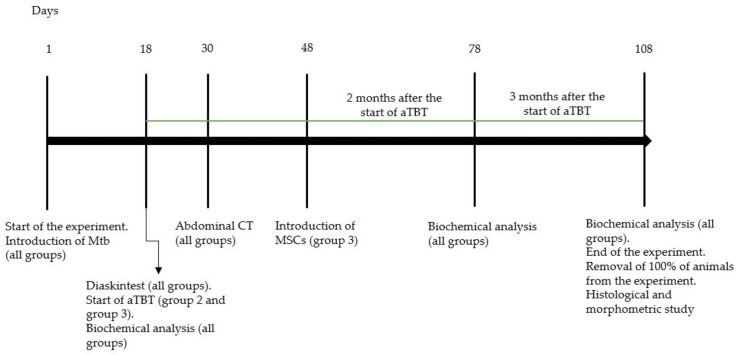
Experimental study design. Abbreviations: Mtb—M. tuberculosis. CT—computed tomography.

**Figure 2 biomedicines-10-03062-f002:**
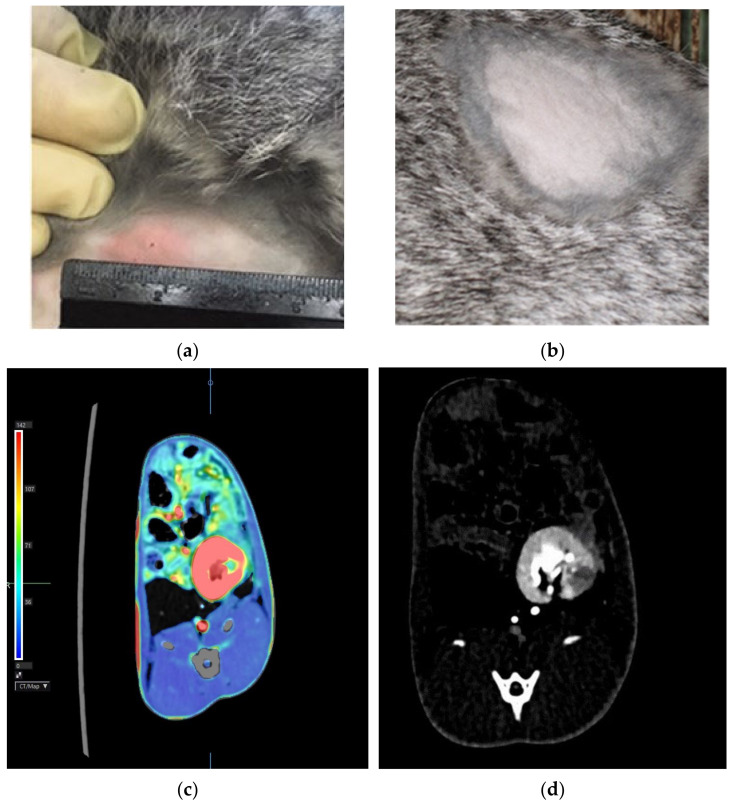
Hyperergic erythema type reaction in response to intradermal administration of Diaskintest 30 days after the challenge (**a**)—infected rabbit; (**b**)—intact rabbit). Abdominal CT: (**c**)—a clearly defined area of reduced perfusion is determined in the left kidney (green color reflects the low blood flow compared to the remaining parenchyma (red)); (**d**)—the destruction focus (the hypovascular area) in the area during standard computed tomography.

**Figure 3 biomedicines-10-03062-f003:**
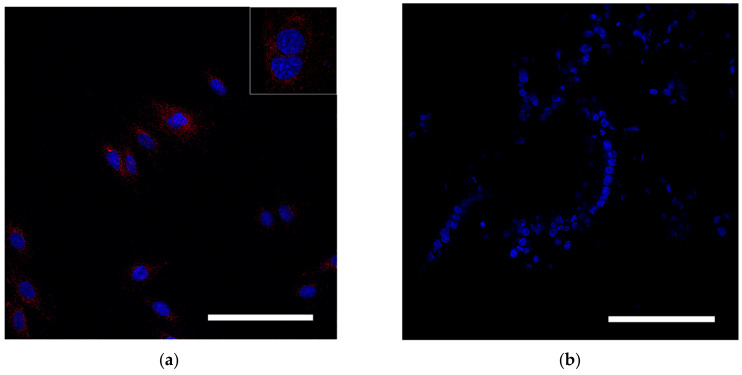
Assessment of SPION internalization with rabbit mesenchymal stem cells. (**a**,**b**) Confocal microscopy images of MSCs co-incubated with SPIONs (150 μg/mL) for 24 h and control (non-treated cells), respectively. Nuclei were stained with DAPI (blue). SPIONs were detected by reflected laser scanning (red). Scale bars, 100 µm.

**Figure 4 biomedicines-10-03062-f004:**
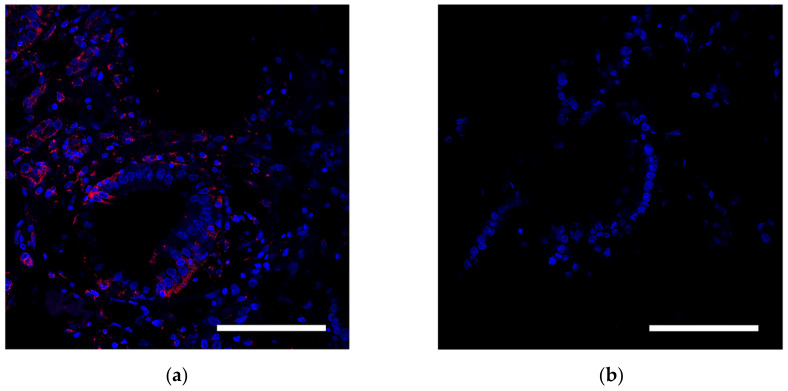
Histological analysis of the kidney. (**a**) Assessment of the SPION-labeled MCS accumulation in the kidney. Nuclei were stained with DAPI (blue). SPIONs were detected by reflected laser scanning (red). (**b**) Control (healthy rabbit without SPIONs administration. Scale bars, 100 µm.

**Figure 5 biomedicines-10-03062-f005:**
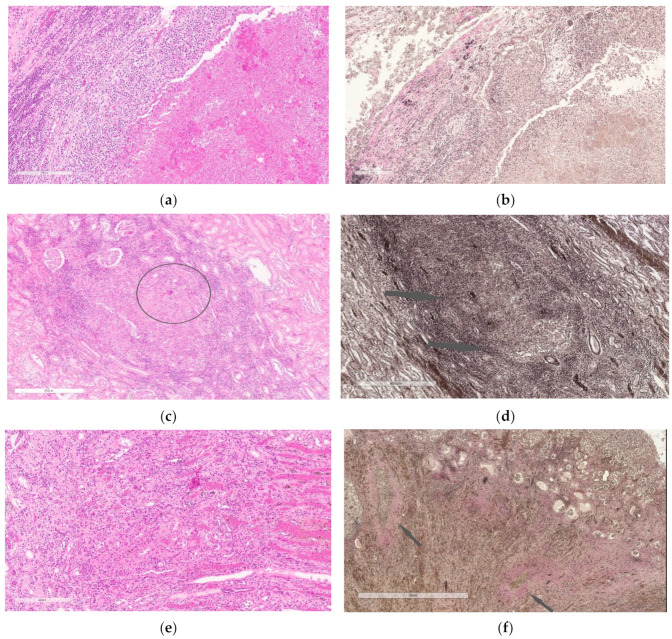
(**a**,**b**)—Kidney of the first group rabbit 3 months after the challenge. A large focus of destruction with a developing three-layer wall and a focus of specific infiltration with necrosis in the center (**a**,**b**). Staining with hematoxylin and eosin (**a**), Van Gieson (**b**). Scale bar, 200 µm (**a**). Scale bar, 200 µm (b). (**c**,**d**)—Kidney of the second group rabbit, which received anti-tubercular chemotherapy within 3 months after the challenge. Focus of specific inflammation with slight necrosis in the center (highlighted circle) and moderately pronounced lymphoid infiltration along the periphery (**c**); mature connective tissue is absent, initial signs of connective tissue formation (indicated by arrows) (**d**). Staining with hematoxylin and eosin (**c**), Van Gieson (**d**). Scale bar, 200 µm (**c**). Scale bar, 400 µm (**d**). (**e**,**f**)—Kidney of the third group rabbit, which received MSCs after completion of 3-month anti-tubercular chemotherapy. Granulation tissue with signs of maturation (**e**), foci of mature connective tissue (indicated by arrows) with accumulations of macrophages in the center of the foci (**f**). Staining with hematoxylin and eosin (**e**), Van Gieson (**f**). Scale bar, 200 µm (**e**). Scale bar, 200 µm (**f**).

**Figure 6 biomedicines-10-03062-f006:**
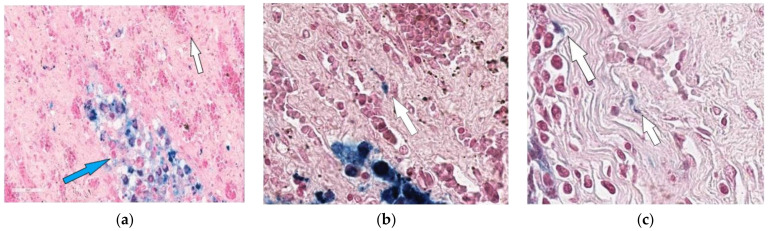
Kidney of the rabbit that received MSCs after completion of 3-month anti-tubercular chemotherapy (one month after MSC transplantation). Accumulation of hemosiderophages (indicated by blue arrow) in the renal parenchyma (**a**) and cells with Perls-positive inclusions vessels (indicated by white arrow) in the walls of small (**b**,**c**). 400×.

**Figure 7 biomedicines-10-03062-f007:**
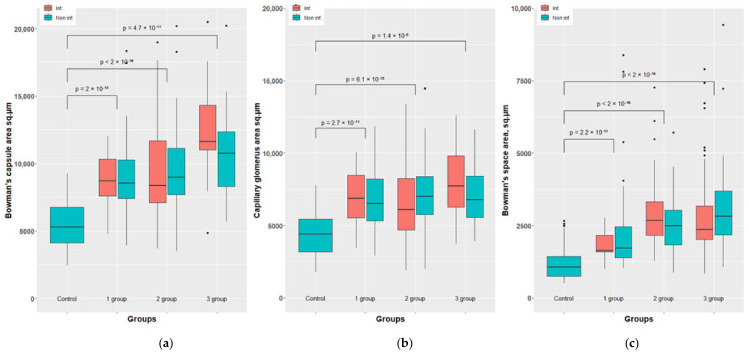
Morphometric results of the rabbit’s kidney of groups 1-3 vs. control. Bowman’s capsule area sq.µm (**a**). Capillary glomerus area sq.µm (**b**) and Bowman’s space area sq.µm (**c**).

**Figure 8 biomedicines-10-03062-f008:**
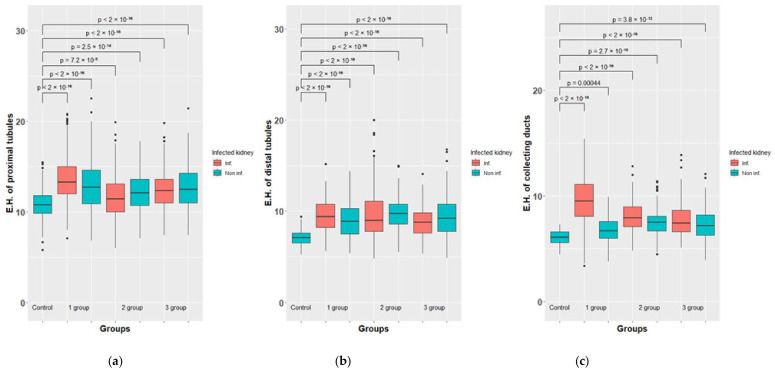
Morphometric results of the rabbit’s kidney of groups 1-3 vs. control. The height of epitheliocytes (E.H.) of proximal tubules, µm. (**a**). The height of epitheliocytes (E.H.) µm. of distal tubules (**b**) and collecting ducts (**c**).

**Table 1 biomedicines-10-03062-t001:** Biochemical indicators in the studied groups.

Indicators	Baseline Values	Examination Groups								
		1			2			3		
		18 Days	2 Months	3 Months	18 Days	2 Months	3 Months	18 Days	2 Months	3 Months
CR Creatinine μmol/L	67.0 (60.5; 73.5)	66.0 (59.0; 80.0)	89.8 * ^(*p* = 0.019)^ (76.0; 98.0)	84.0 * ^(*p* = 0.006)^ (75.0; 91.0)	66.0 (59.0; 80.0)	73.0 (64.0; 81.0)	77.0 * ^(*p* = 0.01)^ (73.0; 93.0)	66.0 (59.0; 80.0)	73.0 (64.0; 81.0)	74.0 * ^(*p* = 0.049)^ (71.0; 79.0)
CP g/L	0.25 (0.16; 0.34)	0.32 (0.24; 0.38)	0.36 (0.29; 0.52)	0.38 (0.29; 0.58)	0.32 (0.24; 0.38)	0.28 (0.23; 0.38)	0.23 **^ (*p* = 0.026)^ ^*** (*p* = 0.008)^ (0.2; 0.25)	0.32 (0.24; 0.38)	0.28 (0.23; 0.38)	0.21 **^ (*p* = 0.01)^, ^*** (*p* = 0.02)^ (0.18; 0.29)
Al Albumin g/L	45.7 (43.5; 46.0)	47.0 (45.0; 49.0)	46.0 (20.6; 48.0)	46.0 (43.0; 49.0)	47.0 (45.0; 49.0)	36.0 ** ^(*p* = 0.03)^ (19.3; 48.0)	46.0 (45.0; 47.0)	47.0 (45.0; 49.0)	36.0 ** ^(*p* = 0.03)^ (19.3; 48.0)	48.0 * ^(*p* = 0.03)^ (45.0; 51.0)
Total ADA U/L	3.8 (2.3; 5.8)	8.4 * ^(*p* = 0.0015)^ (5.2; 12.6)	15.6 * ^(*p* = 0.003)^ (11.1; 18.8)	9.9 (5.8; 11.1)	8.4 * ^(*p* = 0.001)^ (5.2; 12.6)	8.5 * ^(*p* = 0.01)^ *** ^(*p* = 0.025)^ (5.3; 11.7)	13.7 * ^(*p* = 0.01)^ (10.9; 15.1)	8.4 * ^(*p* = 0.0015)^ (5.2; 12.6)	8.5 * ^(*p* = 0.01)^( *** ^(*p* = 0.025)^ 5.3; 11.7)	15.0 * ^(*p* = 0.016)^ (4.6; 17.8)
Ecto-ADA-1 U/L	3.3 (1.9; 5.2)	8.4 * ^(*p* = 0.005)^ (4.4; 12.5)	12.6 * ^(*p* = 0.005)^ (9.7; 16.6)	9.9 * ^(*p* = 0.027)^ (5.8; 10.6)	8.4 * ^(*p* = 0.005)^ (4.4; 12.5)	7.7 * (*^p^* ^= 0.02),^ ^*** (*p* = 0.03)^ (4.3; 11.7)	11.7 * ^(*p* = 0.002)^ (11.5; 14.9)	8.4 * ^(*p* = 0.005)^ (4.4; 12.5)	7.7 * ^(*p* = 0.02)^ *** ^(*p* = 0.03)^ (4.3; 11.7)	13.3 * ^(*p* = 0.01)^ (4.5; 14.8)
Ecto-ADA-2 U/L	0.47 (0; 1.3)	0.02 (0; 0.95)	1.92 (0; 2.3)	0.2 (0; 0.45)	0.02 (0; 0.95)	0 (0; 0.8)	0.1 (0; 1.6)	0.02 (0; 0.95)	0 (0; 0.8)	1.3 (0; 2.1)
El Elastase IU	326.0 (211.9; 393.0)	445.5 * ^(*p* = 0.049)^ (282.5; 510.7)	456.4 (380.3; 500.0)	391.1 (260.8; 423.8)	445.5 * ^(*p* = 0.049)^ (282.5; 510.7)	358.6 (298.9; 489.0)	385.6 (315.0; 434.7)	445.5 * ^(*p* = 0.049)^ (282.5; 510.7)	358.6 (298.9; 489.0)	326.0 (315.0; 358.6)
TIMP-1. ng/mL	103.0 (79.1; 128.9)	84.5 * ^(*p* = 0.03)^ (70.3; 113.9)	107.5 * ^(*p* = 0.03)^ (83.8; 108.3)	98.9 (76.4; 106.4)	84.5 * ^(*p* = 0.04)^ (70.3; 113.9)	91.2 (78.6; 104.7)	90.4 * ^(*p* = 0.04)^ (73.7; 104.0)	84.5 * ^(*p* = 0.04)^ (70.3; 113.9)	91.2 (78.6; 104.7)	94.4 (87.8; 109.5)
MMP-1: ng/mL	0.39 (0.35; 0.42)	0.36 (0.33; 0.43)	0.39 (0.34; 0.393)	0.33 * ^(*p* = 0.03)^ (0.32; 0.47)	0.36 (0.33; 0.43)	0.39 (0.34; 0.46)	0.41 (0.34; 0.56)	0.36 (0.33; 0.43)	0.39 (0.34; 0.46)	0.38 (0.35; 0.42)
MMP-3: ng/mL	20.7 (6.3; 27.3)	21.6 (11.6; 23.8)	13.4 (7.3; 19.8)	7.1 (3.6; 10.7)	21.6 (11.6; 23.8)	9.2 (1.2; 22.2)	5.0 (0.1; 23.1)	21.6 (11.6; 23.8)	9.2 (1.2; 22.2)	10.6 (6.7; 15.8)

Note. CR—creatinine concentration, AL—albumin concentration, CP—ceruloplasmin concentration, Total ADA—adenosine deaminase activity, ecto-ADA-2, EL—elastase activity, TIMP-1—tissue inhibitor 1 concentration, MMP-1—ecto-1 concentration, MMP-3—concentration of metalloproteinase-3, * statistically significant difference compared to the baseline; ** statistically significant difference compared to 18 days *** significant difference from the first group.

## Data Availability

Not applicable.

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
