# Peer review of "The Use of Mesenchymal Stem Cells in the Complex Treatment of Kidney Tuberculosis (Experimental Study)"

_biomedicines, 2022, doi:10.3390/biomedicines10123062_

Round 1
Reviewer 1 Report
This manuscript is so organized, and I think this is almost enough for acceptance. However, there are some mistakes or insufficient points. Please revise them as following comments:
Minor comments
1. Text
1) Figure legends: almost all figure legends have insufficient descriptions and we cannot understand exactly each figure. Please check them again and change them to a more detailed description.
2) All text: there are a lot of spelling mistakes or insufficient word spaces. Please check all the text and revise the inappropriate sentences.
3) Title page: For the author names, the numbers of affiliation are not superscript. Please revise them.
2. Figures
1) Figures 1c/1d/2/3: I think they are unclear photos. Please change them to clearer ones.
2) Figure 4: I think the authors should have shown the result of Van Gieson staining on 1st group.
3) Figure 5: I think these photos are on 3rd group only. However, I think the data on the 1st and 2nd groups are necessary. Please the data on add 1st/2nd group.
Author Response
Response Letter
Dear Editorial Office,
We are grateful for the provided comments and propose the revision of our manuscript. According to the reviewers comments we have revised our manuscript and have incorporated the changes.
The answers to the reviewers are written below.
Reviewers' comments:
Reviewing: 1
Minor comments
- Text
1) Figure legends: almost all figure legends have insufficient descriptions and we cannot understand exactly each figure. Please check them again and change them to a more detailed description.
ANSWER: The numbers of figures and their legends (2,5,6) were corrected.
2) All text: there are a lot of spelling mistakes or insufficient word spaces. Please check all the text and revise the inappropriate sentences.
ANSWER: The manuscript was checked by our institute translator.
3) Title page: For the author names, the numbers of affiliation are not superscript. Please revise them.
ANSWER: The numbers of affiliation were revised and corrected.
- Figures
1) Figures 1c/1d/2/3: I think they are unclear photos. Please change them to clearer ones.
ANSWER: Figures 1c,1d, 2, 3 were uploaded in better quality.
2) Figure 4: I think the authors should have shown the result of Van Gieson staining on 1st group.
ANSWER: The result of Van Gieson staining on 1st group was shown.
3) Figure 5: I think these photos are on 3rd group only. However, I think the data on the 1st and 2nd groups are necessary. Please the data on add 1st/2nd group.
ANSWER: Histological examination in the first group revealed necrosis and severe dyscirculatory disorders (therefore it was not necessary to use additional method). As for the second group, histological examination clearly visualized hemosiderophages in the areas of inflammation, which did not require the use of additional staining. We used this type of staining as an additional method for MSCs detection in kidney tissues.

Reviewer 2 Report
Thank you for the opportunity to review this manuscript which aimed to evaluate the effectiveness MSCs in the activation of reparative 89 processes on the bladder and female genitalia TB model.
Findings showed that the level of proteins of the inflammatory reaction activity decreased, the area of specific and destructive inflammation in kidneys decreased, and the formation of mature connective tissue was noted, which indicates the reparative reaction activation.
Line line 55 - Write in full the acronym MSC
Material e Methods
I suggest drawing an experimental design of the study.
Line 96 - included entering the weight of the animals
Line 105 - Better describe the obtained cells in the animal.
Results
Lines 328 - 330 - In the figures, the morphology by hematoxylin and others changes the color of the scale better visualize.
Author Response
Response Letter
Dear Editorial Office,
We are grateful for the provided comments and propose the revision of our manuscript. According to the reviewers comments we have revised our manuscript and have incorporated the changes.
The answers to the reviewers are written below.
Reviewers' comments:
Reviewing: 2
Findings showed that the level of proteins of the inflammatory reaction activity decreased, the area of specific and destructive inflammation in kidneys decreased, and the formation of mature connective tissue was noted, which indicates the reparative reaction activation.
Line line 55 - Write in full the acronym MSC
ANSWER: Acronym MSC (mesenchymal stem cells) was added.
Material and Methods
I suggest drawing an experimental design of the study.
ANSWER: The experimental design of our study was drawn.
Line 96 - included entering the weight of the animals.
ANSWER: The weight of the animals was included.
Line 105 - Better describe the obtained cells in the animal.
ANSWER: The bone marrow was collected from healthy chinchilla rabbits (n=4). For all four rabbits, 8 ml of bone marrow was aspirated aseptically from the iliac crest: 4 vacutainers (volume 2 ml) with lithium heparin (BD Biosciences, France) were taken from each animal. MSCs were obtained using ficoll gradient method. The bone marrow was diluted (1:1) with PBS (phosphate saline buffer) and transferred in the 50 ml conical centrifuge tubes (TPP, Switzerland). The tubes SetMate (Stemcell Technologies, USA) and ficoll (Merck, USA) were used for cell isolation. The bone marrow was gently added to the ficoll surface with proportion 2:1. The centrifugation was performed for 10 min at 1200 g. After centrifugation plasma with mononuclear cells were carefully transferred in the new 50 ml centrifugation tube and washed twice with 10 ml PBS. After washing, cells were counted and plated into the dish (Nunc, USA) in the density 5*103/cm2.
Results
Lines 328 - 330 - In the figures, the morphology by hematoxylin and others changes the color of the scale better visualize.
ANSWER: The figures were corrected.

Round 2
Reviewer 1 Report
Thank you for your polite responses and revisions.
I checked them and I have agreed with your responses, and so I think this manuscript can be acceptable.